# Phytochemicals Recovery from Grape Pomace: Extraction Improvement and Chemometric Study

**DOI:** 10.3390/foods12050959

**Published:** 2023-02-24

**Authors:** Maura Ferri, Vasco Lima, Alessandro Zappi, Ana Luísa Fernando, Dora Melucci, Annalisa Tassoni

**Affiliations:** 1Department of Biological, Geological and Environmental Sciences, University of Bologna, 40126 Bologna, Italy; 2LAQV-REQUIMTE, Department of Chemistry, University of Aveiro, 3810-193 Aveiro, Portugal; 3Department of Chemistry “Giacomo Ciamician”, University of Bologna, 40126 Bologna, Italy; 4MEtRICs, CubicB, Departamento de Química, NOVA School of Science and Technology, FCT NOVA, Universidade Nova de Lisboa, Campus de Caparica, 2829-516 Caparica, Portugal; 5Interdepartmental Centre of Agri-Food Industrial Research, University of Bologna, 47521 Cesena, Italy

**Keywords:** by-products, chemometrics, enzymatic extraction, Garganega, grape pomace, Merlot, phenolic compounds, organic solvents, RP-HPLC-DAD

## Abstract

In the last 20 years, an increased interest has been shown in the application of different types and combinations of enzymes to obtain phenolic extracts from grape pomace in order to maximize its valorization. Within this framework, the present study aims at improving the recovery of phenolic compounds from Merlot and Garganega pomace and at contributing to the scientific background of enzyme-assisted extraction. Five commercial cellulolytic enzymes were tested in different conditions. Phenolic compound extraction yields were analyzed via a Design of Experiments (DoE) methodology and a second extraction step with acetone was sequentially added. According to DoE, 2% *w/w* enzyme/substrate ratio was more effective than 1%, allowing a higher total phenol recovery, while the effect of incubation time (2 or 4 h) variation was more enzyme-dependent. Extracts were characterized via spectrophotometric and HPLC-DAD analyses. The results proved that enzymatic and acetone Merlot and Garganega pomace extracts were complex mixtures of compounds. The use of different cellulolytic enzymes led to different extract compositions, as demonstrated using PCA models. The enzyme effects were observed both in water enzymatic and in the subsequent acetone extracts, probably due to their specific grape cell wall degradation and leading to the recovery of different molecule arrays.

## 1. Introduction

Grape is one of the most cultivated crops in the world with an estimated production in 2020 of more than 78 million tons, especially in China, Italy, Spain, France and the United States of America [1]. In its statistical report, the Organization of Vine and Wine reported that 77.8 million tons of grape were produced in 2018 of which 57% was of wine grape, 36% of table grape and 7% of dried grape [2].

Grapes are very interesting from a nutritional point of view, due to their vitamin and mineral contents, and are also rich in numerous phenolic compounds that are distributed in the pulp (10%), seeds (60–70%) and skin (28–35%). These compounds are mainly responsible for wine color, taste, mouth feel and oxidation level [3]. Winemaking is a seasonal activity that, yearly and for a short period, produces large quantities of waste (e.g., 10 million tons in 2018) [4], causing a serious environmental problem [5]. Winery by-products represent more than 30% of the grapes used for winemaking [6], among which a large amount of wastewater and organic wastes are generated [7]. The most abundant winemaking by-product is grape pomace (GP), which consists mainly of pressed skins, seeds and stems [8,9,10]. It is estimated that 1 kg of GP is generated for each 6 L of wine [11].

GP can be disposed as natural waste or used as compost or animal feed, even though its phenolic content could be a limiting factor for those applications as it may inhibit plant germination or reduce feed digestibility [12]. Moreover, the disposal option could be environmentally detrimental given the presence of phenolic compounds that decrease the GP pH while increasing its resistance to biological degradation [13]. GP can be also used to extract tartaric acid, to produce ethanol, biochar, biopolymers and composites [10,12].

More sustainable seems to be the extraction of GP bioactive compounds, which could also be more economically viable than other valorization options allowing the recovery of valuable phenolic compounds, dietary fibers and oils [10]. The spectrum of extractable compounds may vary according to the differences between red and white winemaking processes [8,14,15]. Anthocyanins are the main polyphenolic compounds in red grapes while flavanols are more abundant in white grape varieties [16]. Amongst the GP components, qualitative and quantitative variations in phenolic composition can be detected [17]: Skins are rich in anthocyanins and flavonols [16,18], seeds are rich in procyanidins [19] and flavanols such as catechin and epicatechin [16,17], stems are rich in tannins [20] and flavanols [17]. After winemaking, considerable amounts of such bioactive molecules remain in GP vegetable tissues [11], making their extraction an attractive, sustainable and cost-effective option for food, cosmetic and pharmaceutical industries. Several GP phenolics showed beneficial activities like antioxidant, inhibition of cancer cell proliferation, anti-inflammatory and anti-cholesterol properties, or were used as natural food colorants and additives for functional foods [12].

The extraction of phenolics from GP is hampered by plant cell walls which may entrap such compounds (present either in free or bound forms), making the hydrolyzation/disruption of the cell wall a compulsory step [21]. Consequently, the selection of the extraction method is very important as it may greatly affect the yield and type of the isolated compounds [22]. Common processes include liquid-liquid and solid-liquid extractions and the use of solvents such as ethanol, methanol or acetone. The extraction efficiency can be optimized through the testing of different solvents, GP particle size, incubation temperature and time [13,22]. For example, a lower particle size increases the superficial area available for mass transfer [23], while moderate temperatures should be used since several phenolic compounds are heat sensitive [19]. More recently, greener processes have also been applied, such as natural deep eutectic solvents (NADES) [24], supercritical fluids (SFE) [25] or pressurized liquid (PLE) [9,26] extractions.

In addition, the use of enzymes (single or in combination) could be a valuable resource to optimize the extraction process since they can enhance the release of the phenolic compounds bound to the GP cell structures, increasing the extraction yield. Cellulases, pectinases, tannases, glucoamylases and proteases are the most used enzymes in the recovery of bioactive compounds in grape by-products [4]. Although the use of enzymes for the isolation of phenolics is not innovative, the number of studies focusing on the functional characteristics and industrial applicability of the extracts obtained after enzymatic hydrolysis has been largely growing in the last 20 years, also because this extraction strategy is considered technologically promising and green [4]. With respect to phenolic compounds, the enzyme-assisted extraction consists of the hydrolysis of cell wall macromolecules in order to release crosslinked molecules, and in the release of phenolics stored in the vacuoles. The adoption of this technology presents several advantages such as avoided use of toxic solvents, reduced extraction time, extracts obtained through environmentally friendly methods [4]. The new trends in the use of enzymes for the recovery of phenols from grape byproducts were recently reviewed [4], and the recovery of GP compounds using enzymatic hydrolysis was considered in line with the industry needs of new low-cost technologies and profit maximization, and with the consumer demand for increasingly sustainable products. The study pointed out decreased enzyme costs for industrial-scale processes with respect to lab-scale processes and the possible use of immobilized enzymes that can be recovered from the extract and reused, making the process cheaper [4].

The objective of this work was to set up a new phenol extraction protocol (by combining enzyme and solvent-base methods) to be applied to white and red GP, aiming at higher extraction yields. Five commercial cellulolytic enzymes (Pentopan Mono BG, Pectinex Ultra SPL, Celluclast, Driselase, Viscozyme) and 75% (*v/v*) acetone were tested. A Design of Experiments (DoE) procedure was carried out in order to find the best extraction conditions and minimize the reagent and energy use without decreasing the extraction yield. To the best of our knowledge, there is no study that evaluates the effect of these different enzymes in a combined water-enzyme assisted extraction followed by an acetone extraction and on the bioactive compounds extracts composition. The results on the different recovered classes of compounds were further elaborated by PCA and differences among the extraction performances and effects of different enzymes evaluated. The collected results have increased the scientific background supporting a compound-specific enzyme-assisted extraction in view of a more targeted scale-up and industrial exploitation. Finally, the work contributes to the targets of the United Nations Sustainable Development Goals (https://sdgs.un.org/, accessed on 9 October 2022), and in particular to Goal 12 (Responsible Consumption and Reduction), which focuses on reducing waste generation through prevention, reduction, recycling and reuse.

## 2. Materials and Methods

### 2.1. Grape Pomace

Two types of grape pomace (GP, *Vitis vinifera* L.) were provided by InnovEn Srl (Verona, Italy), obtained using vinification of a red Merlot cultivar or a white Garganega cultivar. According to winemaking procedures, Merlot GP was collected after pressing and wine fermentation (red wine vinification), while Garganega GP was collected after only softly pressing the grape (white wine vinification specific for Prosecco production). Both GP types were frozen and stored at −20 °C on the same day as wine production and contained berry skins, seeds, petioles and stalks.

The GP dry weight (DW), determined by weighing aliquots of 3 g of fresh weight (FW) placed at 80°C for 48 h, was about 37.8% and 31.8% of the FW for red and white GP, respectively, with a consequent humidity content of 62.2% in Merlot GP and of 68.2% in Garganega GP.

### 2.2. Phenolic Compounds Extraction

GP was ground in a kitchen blender before the extraction. Aliquots of 3 gFW were rehydrated with 30 mL of distilled water (solid:liquid ratio = 1:10), pH was adjusted to 4.5 and the proper amount of enzyme was added. Five different commercial enzymes were tested at enzyme/substrate ratios of 1% and 2% (*w/w*, respectively): Pentopan Mono BG (xylanase from *Aspergillus oryzae*, 2500 U/mL), Pectinex Ultra SPL (pectinase from *Aspergillus aculeatus*, 3800 U/mL), Celluclast (cellulase from *Trichoderma reesei*, 700 U/g), Driselase (from *Basidiomycetes* sp., protein ≥10% by Biuret) and Viscozyme (cellulolytic enzyme mixture from *Aspergillus* sp.), all from Sigma-Aldrich (Milan, Italy). According to the manufacturer’s descriptions, Pentopan is an endo-β-(1→4)-xylanase; Pectinex is a blend of pectinases, hemicellulases and β-glucanases; Celluclast is a cellulase; Driselase is a natural mixture of laminarinase, xylanase, and cellulase; and Viscozyme is a broad-spectrum cellulolytic enzyme mixture.

The enzymatic treatments were incubated at 50 °C in an orbital shaker (150 rpm) for 2 or 4 h (10 treatments and 1 control for each incubation time and for each GP type, 44 tests in total). Enzymatic reactions were stopped by boiling the samples for 10 min. Controls without the addition of enzymes were also performed under the same conditions. After centrifugation (10 min, 4500× *g*, room temperature), the supernatant (enzymatic extract) and pellet fractions were collected and stored at −20 °C.

The pellets from the 2 h treatments with 2% *w/w* enzyme/substrate ratio (five tests and one control for each GP, 12 in total), selected as best conditions, were successively re-extracted for 2 h with 15 mL of 75% (*v/v*) acetone at 50 °C (additional 12 extracts were obtained). The supernatants were separated from the pellets after centrifugation (10 min, 4500× *g*, room temperature), and stored at −20 °C until further analyses.

### 2.3. Extracts Characterisation

Total phenols content was quantified in all the liquid extracts using the spectrophotometric Folin–Ciocalteu assay [27,28]. Phytochemical profiles of enzymatic (2% *w/w* enzyme, 2 h incubation) and acetone extracts and of controls were characterized for total contents of major phenol families using spectrophotometric assays: flavonoids [28,29], flavanols [30], tannins [14,31] and anthocyanins [8]. Protein [32] and reducing sugar contents were also measured [33]. Appropriate dose–response calibration curves were plotted (i.e., gallic acid (GA, 0–15 mg) for total phenols, catechin (CAT) for flavonoids (2–14 mg) and flavanols (1–50 mg), bovine serum albumin (BSA, 0–200 µg) for proteins, glucose (GLUC, 50–500 mg) for sugars) and results were expressed as mg of standard compound equivalents per g of pomace DW. Tannins were expressed in mg/gDW by conversion to cyanidin [31], while anthocyanin results were converted from absorbance to malvidin-3-glucoside (MALV) equivalents and expressed as mg MALVeq/gDW [34].

Specific phenols were identified and quantified with HPLC-DAD analyses [9] in 1 mL extract aliquots of Merlot 2% Pentopan and 2% Celluclast 2 h extracts, of Garganega 2% Viscozyme and 2% Celluclast 2 h samples, of their relative acetone extracts and of controls. These samples were selected by considering both yields and enzyme prices. When acetone was present, it was evaporated and replaced with water. Phenols were purified and concentrated by means of Strata-X columns (33 mm polymeric reversed phase 60 mg/3 mL, Phenomenex, Bologna, Italy) and analyzed in an HPLC system (column Gemini C18, 5 mm particles, 110 Å, 250 × 4.6 mm; precolumn SecurityGuard Ea; Phenomenex) equipped with an on-line diode array detector (MD-2010, Plus, Jasco Europe, Cremella, Italy). The adopted HPLC-DAD separation procedure allowed the simultaneous identification and quantification of 30 compounds [9]. For each extract, five chromatograms obtained at different wavelengths (270, 285, 305, 323 and 365 nm) were analyzed to determine the concentration of single compounds, depending on their maximum absorbance.

### 2.4. Statistical Analysis

In order to find the best extraction conditions, a design of experiments (DoE) procedure was carried out [35]. This method considered the operational variables (in this case percentage of enzyme, incubation time and type of enzyme) as factors, or independent variables, and the extracted phenolic compound contents as response or dependent variable. Successively, the DoE indicated which experiments should be carried out to better describe the entire experimental domain, (the “region” spanned by the combinations minima and maxima values of the factors) and, once the experiments had been executed, computed a model which can be used to find the best factor combination for further analyses. In this case, the best factor combination was the one that maximizes the yield of extracted phenols. A full-factorial DoE with three factors was applied: Two of them (enzyme/substrate ratio (*ratio*) and incubation time (*time*)) are continuous with two levels (codified as −1 and +1), the third (enzyme type (*type*)) is a categorical variable with five levels. Details about the use and the computations of DoE have been previously described [36]. DoE computations were performed with software CAT [37], based on the R 3.1.2 environment (R Core Team, Vienna, Austria).

All the extractions were repeated at least two times and the experimental data were expressed as mean ± SD. All spectrophotometric assay procedures and HPLC-DAD analyses were performed in duplicate in two technical replicates each. The results are expressed as the mean (*n* = 3) ± SD per gram of dry weight (gDW). The analysis of variance was performed using one-way ANOVA followed by post-hoc corrected two-tailed Tukey– HSD tests with a significance level *p* < 0.05 by using IBM SPSS Statistics for Windows version 24.0.

Results were also elaborated by PCA, in order to evaluate differences among the extraction performances of enzymes and solvents. PCA [38] is a well-known chemometric procedure, which rotates the original variable space to obtain a new high-informative space in which the first two (or three) directions (called principal components, PCs) are, generally, sufficient to describe the full samples and variables behavior. In this way, the 2D scores and loadings plots can be used to evaluate the presence of groups of samples and variables with similar behavior. PCA was carried out with the software R v.4.1.0 (R Core Team, Vienna, Austria).

## 3. Results and Discussion

### 3.1. Extraction and DoE Modelling

Merlot (red) and Garganega (white) grape pomace (GP) were treated with five different commercial cellulolytic enzymes (Pentopan, Pectinex, Celluclast, Driselase and Viscozyme) to release phenolic compounds. Temperature (50 °C), solid/liquid ratio (1:10) and stirring conditions were kept constant, while two key parameters were considered for extraction improvement: enzyme/substrate ratios (1% and 2% *w/w*) and incubation time (2 h and 4 h). In this first set of experiments, only enzymatic treatments were performed. A full-factorial DoE was applied, considering all variable combinations, with 20 experiments carried out for each GP type. Extracted phenols content was used as the DoE response.

The results for all experiments are reported in Figure 1. All the tested enzymatic treatments were able to increase the amount of extracted phenols (up to averages of 40.9% and 24.9% of the samples after the five enzymatic treatments on white and red GP, respectively, 2 h incubation) when compared to the relative control GP samples incubated at the same conditions without enzyme (Figure 1). By extending the incubation time to 4 h, total levels of extracted compounds were slightly increased, but mainly in the controls. Therefore, the effect of the enzymes after 4 h was less incisive, when compared to related controls at 4 h, and the improved yields were, on average among the five enzymatic treatments, only 15.9% and 10.7% for white and red GP, respectively (Figure 1).

Overall, the amounts of total phenols in the enzymatic extracts were on average 26.0 and 25.0 mg GAeq/gDW, for red and white GP, respectively, considering both 2 h and 4 h data (Figure 1). The use of 2% *w/w* enzyme/substrate ratio led to a higher phenol release than 1% *w/w*: on average +24.8% with Merlot after 2 h, +16.4% with Merlot after 4 h and +11.3% with Garganega after 4 h. Only in the case of Garganega treated for 2 h, 1% enzymes were more effective than 2%, on average +24.8% (Figure 1).

In order to evaluate the best extraction conditions, DoE models for red and white GP were computed. Table 1 shows the coefficients estimated for the two models, with star code indicating the statistically significant coefficients (*p* < 0.05). Table 1 clearly indicates that all factors (incubation time, *time*, enzyme/substrate ratio, *ratio*, and enzyme type, *type*) were strongly significant for both red and white pomace extractions. Among the interaction terms, only the one between time and ratio (*time* × *ratio*) was significant for both models, while the interactions of *type* with the other variables (*time* × *type* and *ratio* × *type*) were significant only for the Garganega model. Finally, the square term of *type* (*type*^2^) was not significant for both models, indicating the presence of a linear trend for the variable *type*. The square terms of *time* and *ratio* could not be calculated because only two levels for these factors were considered.

The best extraction conditions could be evaluated by considering the response surface, in which the values of the response were calculated in all experimental domains, between minimum and maximum levels of the factors. *Type* was a categorical variable; therefore, only the values calculated for its five levels (i.e., the five different enzymes) had chemical meaning, the intermediate ones were mathematically computed, but these points could not be chemically reached. Therefore, for each model (Merlot or Garganega), five response surfaces could be obtained, imposing *type* to one of the five levels, and showing the interaction between *time* and *ratio*. The statistical significance of the *type* variable indicated that all considered enzymes were important to improve the phenolic compounds extraction. To allow a better comprehension readability, Figure 2 shows as example the response surfaces (with the related semiamplitudes of the 95% confidence interval, roughly corresponding to the calculated standard deviations, Figure 2b,d) for the Merlot and Garganega GP models at the level of the Pentopan enzyme. The other response surfaces corresponding to the other enzyme-type levels and reporting similar information, were reported as Appendix A.

Figure 2a shows that the best extraction region for red GP (Pentopan enzyme), corresponding to the highest calculated phenol content, was the one with the highest ratio (enzyme/substrate 2% *w/w*). At this ratio, the chosen level for time seemed negligible as the extracted phenol content was the same for both 2 h and 4 h incubation times. This feature was not totally similar for the other enzymes, whose response surfaces (Appendix A) showed some curvature also for the highest time level indicating a slight improvement of extraction at 4 h, in particular for Viscozyme (Appendix A). The semiamplitude (*σ_y_*) plots (Figure 2b,d) indicated that the calculated values in the response plot (Figure 2a,c) were always significantly different from zero, therefore all points were chemically significant (*y ± σ_y_* is always higher than zero, therefore the calculated concentrations have always non-negative values).

In the case of white Garganega GP, Figure 2c shows that the best working region was at the highest levels of both time (4 h) and enzyme/substrate ratio (2% *w/w*) or at low levels of both (2 h and 1% *w/w*). Figure 2d shows that all points were significantly different from zero. The same conclusions could be drawn also for the other enzymes (Appendix A), although, for Viscozyme (Appendix A), the effect of the enzyme ratio factor was slightly lower when combined with the highest time level.

The indications of DoE were used to select the best conditions for the extraction of phenols: minimum time level (2 h) and maximum enzyme/substrate ratio (2% *w/w*). While for red Merlot GP this was still a good choice, because at the highest enzyme/substrate ratio the influence of time was reduced, for white Garganega GP it would have been better to incubate the samples 4 h (Figure 1 and Figure 2). However, higher incubation time would have led to a larger consumption of energy resources (besides obviously increasing the duration of the process) without obtaining, according to the DoE model, a significant gain in extracted phenolic compounds (about 7 mg GAeq/gDW on average for all enzymes) (Figure 2c and Appendix A). Therefore, the 2 h time level was chosen to keep the extraction procedures of red and white GP and their results directly comparable.

Subsequently, solid residues deriving from all enzymatically digested samples (2% *w/w* enzyme, 2 h) were further extracted for an additional 2 h with 75% (*v/v*) acetone.

### 3.2. Extracts Characterization

Both enzymatic (2% *w/w* enzyme, 2 h) and acetone liquid supernatant extracts were then spectrophotometrically characterized for total contents of phenols, flavonoids, flavanols, tannins, anthocyanins, proteins and reducing sugars (Figure 3), and specific phenols were quantified using HPLC-DAD (Table 2).

Higher amounts of total phenols were extracted from Merlot GP with respect to Garganega GP (on average, in enzymatic + acetone extracts, 55.5 and 47.6 mg GAeq/gDW respectively), with almost equal levels in acetone (51%) and enzymatic (49%) samples (Figure 3a). In the case of Merlot, the total level of phenols reached after the two steps extraction, was 11% higher than phenols obtained only with the acetone extraction (under similar conditions) from the same GP [9]. The observed flavonoids trend was similar to total phenols, with only slightly significant differences among the five digestion treatments (Figure 3b). Maximum yields of flavanols were obtained with Celluclast (36.4 and 27.8 mg CATeq/gDW from Merlot and Garganega GP, respectively, Figure 3c). Tannins were extracted mainly from white (20.5 mg/gDW) compared to red GP (15.1 mg/gDW), and by the enzymatic extraction (61%) (Figure 3d). On the contrary, anthocyanins were more abundant in Merlot extracts (on average in total 0.48 mg MALVeq/gDW) with respect to Garganega samples (in total 0.36 mg MALVeq/gDW) (Figure 3e).

Proteins were extracted in significant amounts from both GP with a maximum average of 210.5 mg BSAeq/gDW in Garganega enzymatic extracts (Figure 3f). Previous literature concerning phenolic compound extraction from GP did not generally report data on protein levels. Instead, when GP was treated with proteases a water-soluble extract with anti-inflammatory activity and containing peptides, carbohydrates, lipids and polyphenols (mainly flavonoids and phenolic acids), was obtained [39]. Similarly, data on Figure 3f shows that proteins were co-extracted with phenols and also when cellulolytic enzymes were applied.

Reducing sugars were present on average 4.4-times more in white than in red samples, and in particular in enzymatic extracts (82.2% of total enzymatic + acetone level) (Figure 3g). In Merlot GP, Viscozyme was the enzyme able to release the highest level of sugars (Figure 3g) in agreement with literature, where Viscozyme was found useful for xylo-oligosaccharides and xylose recovery from red Syrah GP [40].

Aqueous and acetone extracts from controls, and water enzymatic and acetone extracts from two enzyme treatments for each GP type (selected by considering both yields (Figure 3) and enzyme prices in view of future industrial exploitation) were analyzed using HPLC-DAD for specific phenols identification and quantification (Table 2).

A wide spectrum of compounds was identified in both Garganega and Merlot GP extracts. Overall, higher concentrations were recovered in white with respect to red samples, and in enzymatic compared to acetone extracts (Table 2).

The most abundant compounds belong to the flavanols family (two isoforms of epigallocatechin and catechin, and one of epicatechin), reaching up to 1.57 mg/gDW of epigallocatechin isoform 2 in Celluclast enzymatic extract of Garganega (Table 2). The present high levels of catechins make the extracts promising for further exploitation. For example, it was demonstrated that a Merlot GP extract rich in catechins and anthocyanins was able to improve the oxidative and inflammatory state of rats with adjuvant-induced arthritis [41].

The second phytochemical family largely present was that of hydroxybenzoic acids, which includes gallic, protocatechuic, vanillic, and syringic acids. These compounds were more abundant in Merlot compared with Garganega GP (in which vanillic and syringic acids were absent and protocatechuic acid was very low) and their extraction was not significantly affected by the type of enzymatic treatment. The only exception was syringic acid, which was increased by 54.5% in Celluclast enzymatic extract of Merlot GP with respect to control (Table 2). These data are in agreement with the findings of Meini [12] that identified syringic acid as the major component of red GP (mixed cultivars) extract with an increase after cellulase treatment. Other identified flavonols were quercetin and its derivative rutin. Quercetin was present only in acetone extracts, while rutin was detected only in Garganega samples (Table 2).

Two stilbenes were also identified: *cis*-piceid and, in lower amount, *cis*-resveratrol. The former was extracted from Merlot GP on average 1.5 times more by the enzymatic step than by acetone, while on the contrary, in Garganega GP it was 2.9 times higher in acetone than in enzymatic samples. *Cis*-resveratrol was present only in aqueous control samples (Table 2). The absence of resveratrol in GP extracts has been previously reported [12].

In general, the yields of most of the identified phenols increased in the combined enzyme plus acetone extractions with respect to those measured in extracts obtained, from the same GP, only with solvents. For example, the most abundant compound previously quantified in acetone extracts (under similar conditions) was catechin 2, with yields of 0.348 and 0.523 mg/gDW for Merlot and Garganega GP respectively [9,26]. In the present results, the reported catechin 2 yields were increased 2.3 and 3.0 times in Merlot GP extracted with Pentopan or Celluclast, plus acetone, and 3.4 and 3.0 times in Garganega GP extracted with Celluclast or Viscozyme, plus acetone (Table 2).

The type and amount of compounds present in the extracts were dependent on several factors. For example, phytochemical profile variations between Merlot and Garganega (Figure 3 and Table 2) can be ascribed to the different grape cultivars, while the higher reducing sugar content of white extracts (Figure 3g) was mainly due to the distinct winemaking techniques applied (Garganega GP was obtained immediately after a soft pressing of the grapes, while Merlot GP was collected at the end of wine fermentation).

### 3.3. PCA Analyses

One of the aims of the present study was to infer if the treatment with different enzymes could also be responsible for diverse extract compositions. Spectrophotometric (Figure 3) and chromatographic (Table 2) results gave some indications and, to deeper investigate the enzyme effects, PCA analyses were carried out.

A first PCA was calculated for both GP types by comparing the data of enzymatic and acetone extracts. Figure 4 shows the scores and loading plots for PCA models of both GP. The scores clearly indicated that the two extraction methods brought different results because the enzymatic and acetone sample clusters were sharply separated in both GP (Figure 4a,c). Interestingly, the loading plots (Figure 4b,d) allowed evaluation of the extraction efficiency of the two solvents. Indeed, a specific correspondence was present between the samples in a certain quadrant of the score plots (Figure 4b,d) and the compounds in the same quadrant of the loading plots (Figure 4a,c). For both GP types, enzymes extracted more proteins, anthocyanins, and tannins, while acetone showed a better performance in the release of flavonoids and flavanols. In white GP samples, phenolic compounds were clearly extracted in higher amounts using acetone while reducing sugars using an enzymatic treatment (Figure 4d). Instead, in the case of red GP samples, both corresponding loadings had a high positive value of PC2, and PC1 values near zero (Figure 4b), indicating that reducing sugars and phenols could be efficiently extracted from Merlot GP pomace both by enzymes and by acetone.

To compare the extracting performances of the five enzymes, two PCA for each GP type were calculated, splitting the obtained results according to data collected on enzymatic and acetone samples (Figure 5 and Figure 6). Figure 5 shows the two PCA models for red Merlot GP, in which each score, corresponding to a specific enzyme, was identified by a different color (Figure 5a,c). The good grouping of data of the four replicates (two biological analyzed in two technical replicates each) of each enzyme treatment (except for one replicate of Viscozyme in acetone sample, Figure 5c) indicated a good reproducibility of the extraction procedures. The most interesting task, however, was to compare scores and loadings for both enzymatic and acetone extractions, to check which method combination could be considered the most appropriate for each compound family. The PCA model of enzymatic water extracts (62.3% of explained variance carried by PC1 and PC2, Figure 5a,b) indicated that Cellucast (high positive values of PC1) extracted flavanols and anthocyanins better; Driselase (negative values of PC2) was more efficient for flavonoids; Pectinex and Pentopan showed a similar behavior (both with high positive values of PC2) and were more suitable for the extraction of proteins and tannins; Viscozyme (high negative values of PC1) was more suitable for total phenols, reducing sugars, and, to a lower extent, for flavonoids and proteins.

While the data (Figure 3) analyzed for PCA of Figure 5a,b were directly dependent on the enzyme hydrolytic action on GP structure and consequent molecule release in the water solution, the acetone data (Figure 3) analyzed in PCA of Figure 5c,d depended both on acetone extraction capacity (equal for all samples) and on the different GP hydrolysis previously performed by the used enzymes.

The PCA acetone model (60.0% of explained variance carried by PC1 and PC2, Figure 5c,d) indicated the effect on compound release by acetone of former enzymatic hydrolysis with Cellucast and Pentopan having similar negative values for PC1 and being more suitable mostly for flavonoids and flavanols; Pectinex and Viscozyme (high positive values of PC1) allowed a higher extraction of anthocyanins, reducing sugars, and, partially, total phenols; Driselase, instead, had scores grouped close to the origin, seemed to be not particularly suitable for the following acetone extraction of any of the considered species.

PCA models for white Garganega GP (Figure 6) showed no clear division of score groups, both in water enzymatic and in acetone solvents. A partial separation was present in the enzymatic PCA model (60.6% of explained variance carried by PC1 and PC2; Figure 6a,b) only for Cellucast (high positive values of PC2) and Pentopan (negative values of PC2), indicating that the first enzyme could be preferred for the extraction of flavanols, reducing sugars and anthocyanins, and the second for tannins. These results indicated that, in general, for Garganega GP and in both extraction conditions (Figure 6c,d), all the five considered enzymes behaved similarly and none of them appeared to be more specific for the extraction of any of the considered chemical species.

Data on the GP extracted compounds presented in Figure 3 and Table 2 and elaborated via PCA (Figure 4, Figure 5 and Figure 6), reflected the effects of the different used enzymes. Enzyme-assisted hydrolysation of cell wall polysaccharides and other macromolecules allows the release of compounds (phenolics, sugars, proteins, etc) linked to cell wall structures, or facilitates the recovery of compounds stored into the vacuole [4]. These released molecules were recovered in water enzymatic extracts (Figure 3, Table 2) and differences among digestates composition (Figure 5c,d and Figure 6c,d) could be ascribed to the diverse enzyme specificities by hydrolysing different cell wall macromolecular components (Methods, Section 2.2). As an example, Viscozyme cellulolytic enzyme mixture released reducing sugars from Merlot GP (Figure 3g and Figure 5) coherently with the proposed xylo-oligosaccharide production activity detected in red Syrah GP [40]. Meini and coworkers [12] related the phenolic compound extraction yields to the Celluclast hydrolysation effect on red GP cellulosic fibres where phenolics are bound.

Most of the published studies about GP enzyme-assisted extraction only considered the enzymatic step, while few researchers have proved that phenol recovery can be further improved with a second step solvent extraction [8,14]. For example, Ferri and colleagues used ethanol to extract phenols, flavonoids and tannins from Sangiovese and Montepulciano GP [8] and from Trebbiano and Verdicchio GP [14] after different enzymatic treatments aimed at achieving bioactive extracts which are useful for food, cosmetic and pharmaceutical applications.

## 4. Conclusions

The present study investigated the extraction of phenolic molecules and other compounds from grape pomace (GP; Merlot and Garganega cultivars) by means of five different cellulolytic enzymes and the effects of such treatments on the additional acetone extraction step. According to DoE (design of experiments) analyses, a higher enzyme/substrate ratio (2% *w/w* in comparison to 1%) was required for a more efficient total phenolic compounds recovery, while incubation time did not have a significant impact and was more enzyme-dependent. Among the assayed enzymes, Celluclast led to higher extraction yields from both GP types, followed by Pentopan Mono BG (for red Merlot GP) and then Viscozyme (for white Garganega GP). PCA models highlighted the differences among extracts. The results and their PCA analysis proved that both Merlot and Garganega pomace extracts were a complex mixture of compounds and that the use of different cellulolytic enzymes can lead to different extract compositions, both in water enzymatic and in acetone extracts. For both GP types, enzymatic solutions extracted more anthocyanins, tannins and proteins, while acetone showed better performances for flavonoids and flavanols. In Garganega GP, phenols were clearly extracted in higher amounts using acetone and reducing sugars using enzymes, while, in the case of Merlot GP, reducing sugars and phenols can both be efficiently extracted using enzymes and acetone.

Most interestingly, for the first time the effects of the used enzymes were observed both in water enzymatic and in the further acetone extracts, due to the specific grape cell wall degradation and consequent possibility to recover different spectra of molecules. Therefore, the present study allowed the evaluation of the specific GP cell wall deconstruction effect also on the second solvent extraction step, while most of the scientific studies concerning GP enzyme-assisted extraction only considered the enzymatic step.

The application of the enzymes can represent an effective way to enhance the extraction yields or even to try to optimize/reduce the use of solvents. A further interesting step could be the hydrolysis of GP by using immobilized enzymes, therefore allowing their recovery and reuse. Improved processing conditions for the isolation of valuable compounds from GP represents an additional revenue for the wine industry, a cheap source of phenolics and other compounds for other type of industries and a further effort toward diminishing food industry environmental burdens.

## Figures and Tables

**Figure 1 foods-12-00959-f001:**
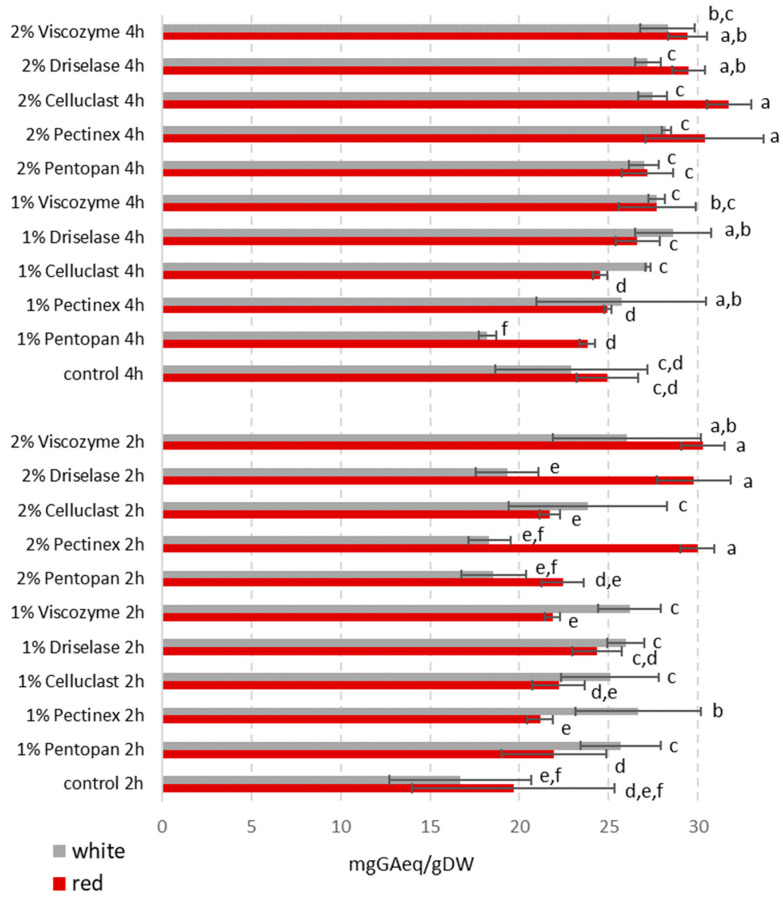
Total phenolic compound contents in white (Garganega) and red (Merlot) grape pomace enzymatic extracts obtained after treatment with five different commercial enzymes at 1% or 2% enzyme/substrate ratios (*w/w*), for 2 h or 4 h. Control samples were subjected to the same treatment conditions without any enzyme. Results are expressed as mg of gallic acid (GA) equivalent per gram of pomace dry weight (DW). Data represent the mean ± SD (*n* = 3). Different letters indicate a statistically significant difference (one-way ANOVA test followed by post-hoc corrected two-tailed Tukey test, *p* < 0.05), from the highest (a) to the lowest (f).

**Figure 2 foods-12-00959-f002:**
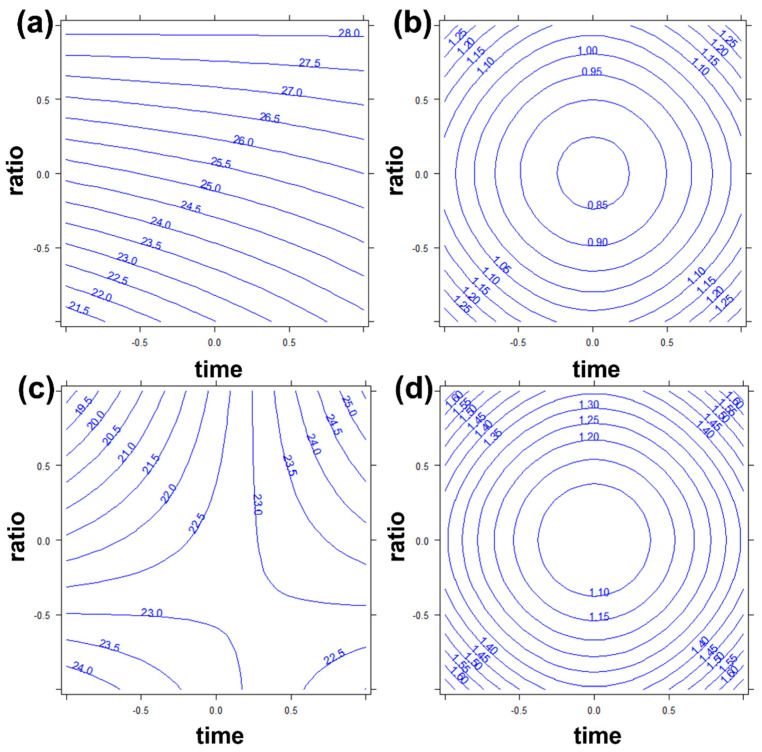
(**a**) Response surface and (**b**) semiamplitude of the 95% confidence interval for Merlot red GP model and (**c**) response surface and (**d**) semiamplitude of the 95% confidence interval for Garganega white GP model. Factors time and ratio are reported in abscissa and ordinate respectively. Enzyme type was kept to Pentopan level. The axis scale ±1 is due to DoE levels.

**Figure 3 foods-12-00959-f003:**
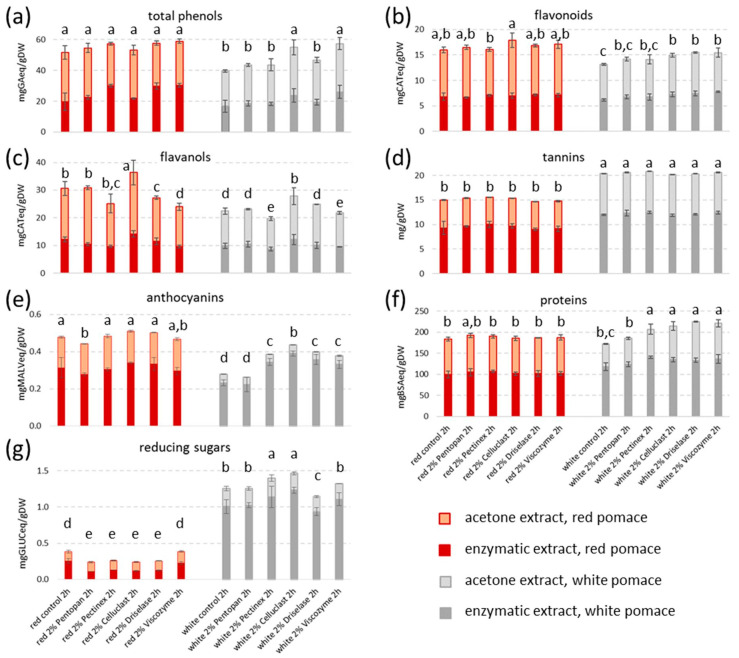
Contents of (**a**) total phenols, (**b**) flavonoids, (**c**) flavanols, (**d**) tannins, (**e**) anthocyanins, (**f**) proteins and (**g**) reducing sugars in red and white GP in enzymatic (2% *w/w* enzyme, 2 h) and acetone (75% *v/v*, 2 h) liquid extracts. Results are expressed as mg of standard compound (GA, gallic acid; CAT, catechin; MALV, malvidin; BSA, bovine serum albumin; GLUC, glucose) equivalents per gram of pomace dry weight (DW). Data represent the mean ± SD (*n* = 3). Different letters indicate a statistically significant difference (one-way ANOVA test followed by post-hoc two-tailed Tukey test, *p* < 0.05) between the same type of data (total phenols, flavonoids, flavanols, tannins, anthocyanins, proteins and reducing sugars), from the highest (a) to the lowest (f). Control: treatment without the addition of enzymes incubated under the same conditions and followed by the same acetone extraction.

**Figure 4 foods-12-00959-f004:**
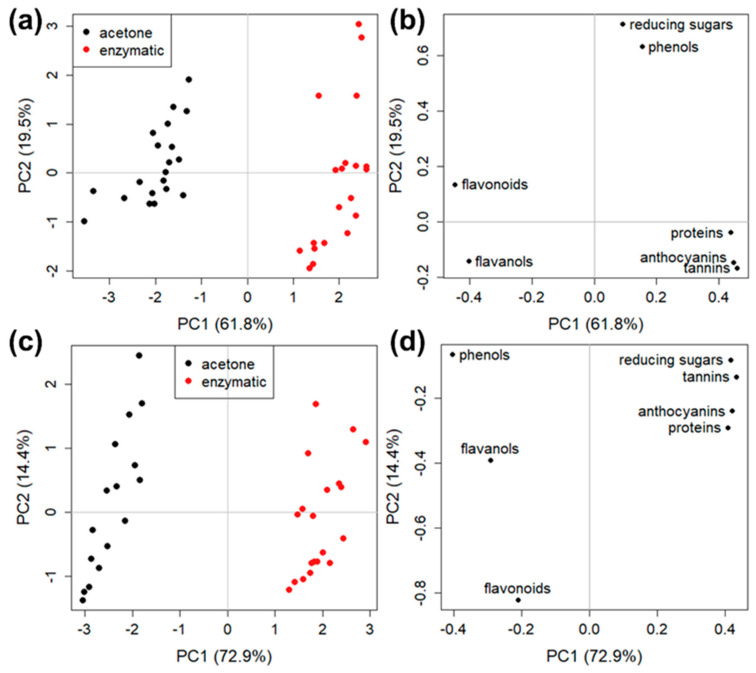
PCA models for (**a**,**b**) Merlot GP and (**c**,**d**) Garganega GP in enzymatic (red points) and acetone (black points) samples; (**a**,**c**) score plots and (**b**,**d**) loading plots.

**Figure 5 foods-12-00959-f005:**
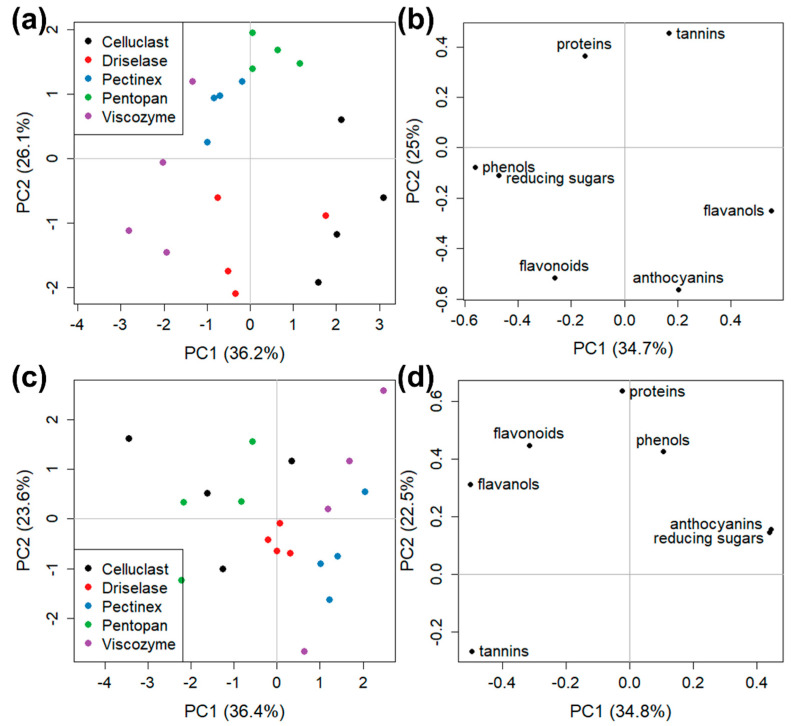
PCA models for (**a**,**b**) enzymatic and (**c**,**d**) acetone Merlot GP extracts. Score colors are based on the different enzymes used for the extraction treatment. (**a**,**c**) score plots and (**b**,**d**) loading plots.

**Figure 6 foods-12-00959-f006:**
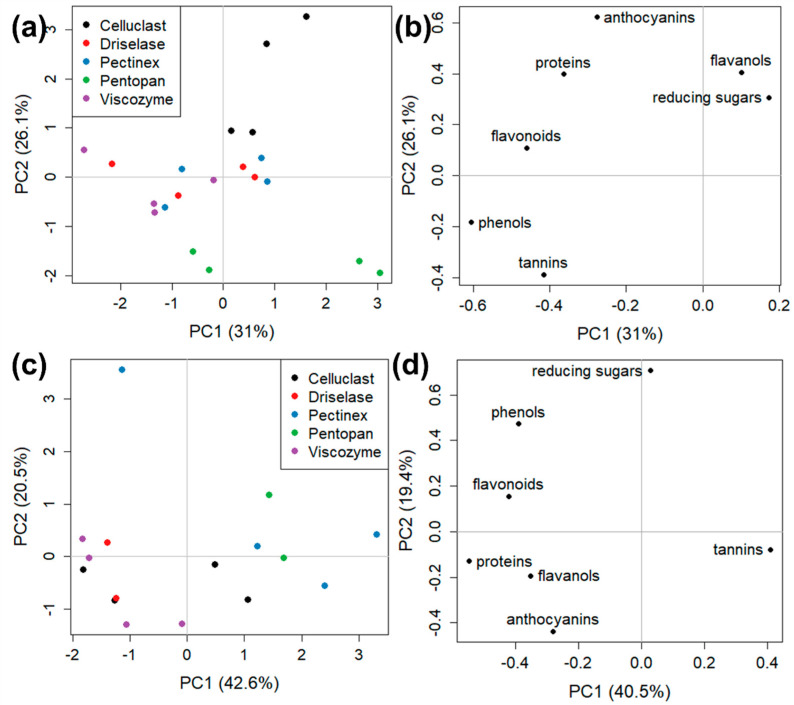
PCA models for (**a**,**b**) enzymatic and (**c**,**d**) acetone Garganega extracts. Score colors are based on the different enzymes used for the extraction treatment. (**a**,**c**) score plots and (**b**,**d**) loading plots.

**Table 1 foods-12-00959-t001:** Regression coefficients estimated for the DoE models. Stars indicate the significance level (* *p* < 0.05; ** *p* < 0.01; *** *p* < 0.001).

	Merlot	Garganega
**b0**	27	24
**time**	0.93 ***	2.1 ***
**ratio**	2.6 ***	−1.2 ***
**type**	1.0 ***	1.4 ***
**time × ratio**	−0.68 ***	2.3 ***
**time × type**	0.27	0.99 **
**ratio × type**	−0.22	−0.72 *
**type^2^**	−0.34	−0.85

b0, intercept; time, incubation time; ratio, enzyme/substrate ratio; type, enzyme type.

**Table 2 foods-12-00959-t002:** Specific phenolic compound levels (mg/gDW) of selected enzymatic and acetone extracts of red and white grape pomace (2 h incubation, 2% *w/w* enzyme/substrate ratio) determined using HPLC-DAD analysis. Results are expressed as mg of compound per gram of pomace dry weight (DW). Data are the mean ± SD of three replicates (*n* = 3). The star symbol (*) indicates a statistically significant difference (one-way ANOVA test followed by post-hoc two-tailed Tukey test, *p* < 0.05) between the enzymatic (water for control samples) or acetone data of a specific treated sample and the respective control.

Compound	Type of Extract	Red Control ^a^	Red Pentopan	Red Celluclast	White Control ^a^	White Celluclast	White Viscozyme
**Epigallocatechin 1**	enzymatic	0.438 ± 0.032	0.430 ± 0.155	0.581 ± 0.126	0.960 ± 0.197	0.980 ± 0.130	0.975 ± 0.035
	acetone	0.201 ± 0.079	0.279 ± 0.026	0.368 ± 0.061 *	0.181 ± 0.066	0.368 ± 0.030 *	0.305 ± 0.049 *
**Epigallocatechin 2**	enzymatic	0.519 ± 0.086	0.530 ± 0.087	0.591 ± 0.038	1.378 ± 0.207	1.570 ± 0.059	1.352 ± 0.164
	acetone	0.122 ± 0.008	0.093 ± 0.010 *	<LoD *	0.226 ± 0.031	0.260 ± 0.029	0.239 ± 0.064
**Catechin 1**	enzymatic	0.249 ± 0.015	0.270 ± 0.049	0.299 ± 0.043	<LoD	0.313 ± 0.014 *	<LoD
	acetone	0.112 ± 0.016	0.083 ± 0.020	0.098 ± 0.021	<LoD	0.065 ± 0.012 *	0.053 ± 0.10 *
**Catechin 2**	enzymatic	0.498 ± 0.045	0.472 ± 0.020	0.777 ± 0.146 *	0.754 ± 0.097	0.856 ± 0.013	0.839 ± 0.138
	acetone	0.708 ± 0.174	0.746 ± 0.044	0.809 ± 0.145	0.594 ± 0.081	0.901 ± 0.084 *	0.729 ± 0.042 *
**Epicatechin**	enzymatic	0.260 ± 0.021	0.390 ± 0.189	0.439 ± 0.084 *	0.271 ± 0.005	0.341 ± 0.034 *	0.297 ± 0.068
	acetone	0.428 ± 0.132	0.451 ± 0.018	0.505 ± 0.088	0.268 ± 0.067	0.355 ± 0.026	0.327 ± 0.033
**Gallic acid**	enzymatic	0.110 ± 0.030	0.128 ± 0.014	0.106 ± 0.069	0.216 ± 0.036	0.253 ± 0.044	0.252 ± 0.042
	acetone	<LoD	<LoD	0.027 ± 0.007 *	<LoD	<LoD	<LoD
**Protocatechuic acid**	enzymatic	0.018 ± 0.001	0.018 ± 0.002	0.021 ± 0.004	<LoD	<LoD	0.017 ± 0.002 *
	acetone	0.007 ± 0.001	0.008 ± 0.001	0.008 ± 0.001	<LoD	0.004 ± 0.001 *	0.004 ± 0.001 *
**Vanilic acid**	enzymatic	0.124 ± 0.004	0.132 ± 0.017	0.156 ± 0.049	<LoD	<LoD	<LoD
	acetone	0.059 ± 0.017	0.060 ± 0.004	0.073 ± 0.009	<LoD	<LoD	<LoD
**Syringic acid**	enzymatic	0.055 ± 0.001	0.059 ± 0.002 *	0.085 ± 0.013 *	<LoD	<LoD	<LoD
	acetone	0.031 ± 0.009	0.026 ± 0.001	0.038 ± 0.003	<LoD	<LoD	<LoD
**Quercetin**	enzymatic	<LoD	<LoD	<LoD	<LoD	<LoD	<LoD
	acetone	0.149 ± 0.006	0.155 ± 0.034	0.210 ± 0.035 *	0.107 ± 0.013	<LoD *	0.118 ± 0.002
**Rutin**	enzymatic	<LoD	<LoD	<LoD	0.424 ± 0.155	0.421 ± 0.043	0.432 ± 0.050
	acetone	<LoD	<LoD	<LoD	0.203 ± 0.010	<LoD	0.205 ± 0.044
***Cis*-piceid**	enzymatic	0.085 ± 0.036	0.095 ± 0.020	0.124 ± 0.045	0.201 ± 0.020	0.191 ± 0.007	0.181 ± 0.057
	acetone	0.136 ± 0.039	0.130 ± 0.002	0.202 ± 0.092	0.057 ± 0.010	0.084 ± 0.010 *	0.050 ± 0.016
***Cis*-resveratrol**	enzymatic	0.014 ± 0.004	<LoD *	<LoD *	0.012± 0.004	<LoD *	<LoD *
	acetone	<LoD	<LoD	<LoD	<LoD	<LoD *	<LoD

^a^ control samples were incubated without the addition of enzymes under the same process conditions; <LoD, below the limit of detection.

## Data Availability

The data presented in this study are available on request from the corresponding author.

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
