# Peer review of "Phytochemicals Recovery from Grape Pomace: Extraction Improvement and Chemometric Study"

_foods, 2023, doi:10.3390/foods12050959_

Round 1

Reviewer 1 Report

The theme is interesting, relevant and a significant number of analyzes were performed, however, there are several aspects to consider:

1.- According to what was reported in the paper, optimization is not achieved.

2.- The methodology is not clear, for example, the number and conditions of the extracts studied, which makes it difficult to interpret the results.

3.- Scarce discussion of the results with those reported in the literature.

4.- Conclude in general, the most relevant aspects found

Author Response

Comments and Suggestions for Authors:

The theme is interesting, relevant and a significant number of analyzes were performed, however, there are several aspects to consider:

1.- According to what was reported in the paper, optimization is not achieved.

The title was modified and the word “optimization” was removed the whole manuscript.

2.- The methodology is not clear, for example, the number and conditions of the extracts studied, which makes it difficult to interpret the results.

Details were added, according to reviewer suggestions into the attached file:

- Introduction section: enzymes and solvent were specified at lines 109-111.

- Materials section 2.1. Grape pomace: humidity percentage was made explicit at lines 140-141.

- Methods section 2.2. Phenolic compounds extraction: the word “respectively” was added at line 146. More details and numbers of performed tests were added (lines 156-157, 161-164). Tannins measure unit was better specified (line 176).

- Results and Discussion section: details were added at lines 230-234, 255-257, 333, 360, 364, 371, 378-379, etc.

Regarding other Reviewer’s comments in the attached file:

- Methods section 2.3: proteins were quantified in the phenolic extracts to assess if proteins were co-extracted together with phenols from grape pomace, therefore we decided to consider their level in characterization analyses and to check if they were treatment-dependent. The following explanation was added in Discussion section (lines 352-358): “Previous literature concerning phenolic compounds extraction from GP, did not generally report data on protein levels. Instead, when GP were treated with proteases a water-soluble extract with anti-inflammatory activity and containing peptides, carbohydrates, lipids and polyphenols (mainly flavonoids and phenolic acids), was obtained [Plant Foods Hum Nutr, 2015, 70: 42-49]. Similarly, data of Figure 3f showed that proteins were co-extracted with phenols also when cellulolytic enzymes were applied.”

- The Reviewer asked why only 2 enzyme treatments for each GP were analyzed by HPLC-DAD. The answer was already present in Results and Discussion section (lines 364-367): “Aqueous and acetone extracts […] selected by considering both yields (Fig. 3) and enzyme prices in view of future industrial exploitation […] (Table 2).” To better clarify why some specific samples were selected, a sentence was added also in Methods section 2.3 (line 182).

- At lines 237-239 and 242-245, the Reviewer asked to clarify % referred to Fig. 1. We calculated those % starting from the data reported into the Figure 1. Considering 100 the phenolic compound content of related 2 h and 4 h controls (made by incubating GP at the same time and conditions but without enzyme) % of increase in each enzymatic extract respectively after 2 or 4 h of incubation was calculated by proportion. Percentages reported into the text are averages among the percentages calculated after the 5 enzymatic treatments of each incubation time and GP type (4 averages: red GP 2h, white GP 2h, red GP 4h, white GP 4h). This is in our opinion a common way to present this type of data and we would prefer not to write all the % into the Figure to avoid the presence of crowded numbers and, therefore, preserve the Figure readability. The text at lines 236-245 was modified to better explain the origin of percentages.

- Moreover, in the attached file, the Reviewer suggested to transform Figure 1 in a Table. In our opinion, a histogram allows an immediate data comparison, showing clearly which treatments led to higher or lower phenolic compounds release and we would prefer to keep the Figure.

3.- Scarce discussion of the results with those reported in the literature.

The Discussion of the new results with those reported in literature was improved and 3 references were added. See lines 352-358, 361-363, 385-389, 396-398, 405.

4.- Conclude in general, the most relevant aspects found

According to specific Reviewer’s comment in the attached file, a sentence was added to Conclusions section. Together with the already presented best process conditions, the enzymes that allowed to the highest compounds yields were indicated. They were Celluclast and Pentopan in case of Merlot pomace and Celluclast and Viscozyme in case of Garganega, according to Figure 3 and Table 2.

English language was also revised and improved.

Reviewer 2 Report

Article: Phytochemicals Recovery from Grape Pomace: Process Optimization and Chemometric Study

In this work the authors propose a novel combination of extraction methodologies of phytochemicals from grape pomace samples as residues of winemaking process/industry, thereby optimizing the extraction process using the design of experiment approach and evaluating the optimized protocol using a chemometric tool, i.e. principal component analysis.

The abstract is clearly written, with a clear representation of the aim of the paper. Furthermore, it is adequately structured: background of the proposed research, methods, results and main conclusions were mentioned.

The title of the paper adequately reflects the subject under investigation in the proposed study.

References are listed numerically, as demanded by formatting rules of the journal. Although there is no limitation in the number of references, a reference list of 38 citations is enough for the topic proposed. Authors mainly use contemporary literature data. However, a high degree of self-citations is observed: the first author 6 self-citations and the last author 7 self-citations.

Throughout the introduction section, the authors show the awareness of the situation in the proposed field.

Line 75: “…and the use OF solvents, such as…

Lines 75-79: Are the authors sure that no greener choices for extraction from grape pomace have been developed, such as supercritical fluids or NADES? Please, mention.

Lines 115-116: The authors mention that “Merlot GP was collected after pressing and wine fermentation, while Garganega GP was collected only after softly pressing the grape.” Could the authors explain the reason for this difference?

Lines 140-141: The authors state that “the pellets from the 2 h treatment with 2% w/w enzyme/substrate ratio were re-extracted with acetone”. There is no explanation why only extracts obtained under these conditions (2 h and 2%) were further used for the extraction with acetone?

Line 510: How cheap is the enzymatic process actually?  The authors are required to insert a reference proving this statement.

Conclusions are adequately supported by the results obtained.

Line 532: Please, separate “by the”.

Generally speaking, the proposed paper in my opinion presents an interesting work and is well-written. 

Author Response

Comments and Suggestions for Authors:

In this work the authors propose a novel combination of extraction methodologies of phytochemicals from grape pomace samples as residues of winemaking process/industry, thereby optimizing the extraction process using the design of experiment approach and evaluating the optimized protocol using a chemometric tool, i.e. principal component analysis.

The abstract is clearly written, with a clear representation of the aim of the paper. Furthermore, it is adequately structured: background of the proposed research, methods, results and main conclusions were mentioned.

Thank you.

The title of the paper adequately reflects the subject under investigation in the proposed study.

Thank you. However, we slightly changed the title according to the Reviewer 1 comments.

References are listed numerically, as demanded by formatting rules of the journal. Although there is no limitation in the number of references, a reference list of 38 citations is enough for the topic proposed. Authors mainly use contemporary literature data. However, a high degree of self-citations is observed: the first author 6 self-citations and the last author 7 self-citations.

Among the 7 self-cited papers, 6 were cited in Methods section proving that the previous experience in such research field. However, 1 self-citation was removed (Ferri et al 2009, former reference n. 32).

Throughout the introduction section, the authors show the awareness of the situation in the proposed field.

Thank you.

Line 75: “…and the use OF solvents, such as…

“Of” was added as suggested.

Lines 75-79: Are the authors sure that no greener choices for extraction from grape pomace have been developed, such as supercritical fluids or NADES? Please, mention.

Green phenols extraction from grape pomace were also considered. A sentence was added in Introduction section and 2 new references were cited (lines 83-85).

Lines 115-116: The authors mention that “Merlot GP was collected after pressing and wine fermentation, while Garganega GP was collected only after softly pressing the grape.” Could the authors explain the reason for this difference?

The different pomace production treatments were consequent of the specific winemaking procedures: red wine vinification for Merlot and white wine vinification specific for Prosecco production for Garganega. Section 2.1 Materials and Methods was improved and specifications added (lines 133-135).

Lines 140-141: The authors state that “the pellets from the 2 h treatment with 2% w/w enzyme/substrate ratio were re-extracted with acetone”. There is no explanation why only extracts obtained under these conditions (2 h and 2%) were further used for the extraction with acetone?

The explanation of such choice was reported into Results and Discussion section. In the last part of section 3.1 Extraction and DoE modelling (lines 318-328), we stated that “The indications of DoE were used to select the better conditions for the extraction of phenols: minimum time level (2 h) and maximum enzyme/substrate ratio (2% w/w). […] Therefore, it was chosen the 2 h time level to keep the extraction procedures of red and white GP and their results directly comparable.”. Acetone extraction was performed only on these selected best enzymatic processes conditions. Method section 2.2 was slightly modified (lines 161-162), while we preferred to leave the extended explanation in the Results and Discussion chapter.

Line 510: How cheap is the enzymatic process actually?  The authors are required to insert a reference proving this statement.

According to reviewer suggestion, we add few lines about enzymes and industrial processes costs and considerations, supported by a recent review (Xavier Machado et al, New trends in the use of enzymes for the recovery of polyphenols in grape byproducts. J Food Biochem 2021, 45, e13712, https://doi.org/10.1111/jfbc.13712). As generally it is not possible to cite any reference in the conclusion section, such paragraph was inserted into the Introduction (lines 100-106).

Conclusions are adequately supported by the results obtained.

Thank you. However, we slightly changed the Conclusions section according to Reviewer 1 comments.

Line 532: Please, separate “by the”.

Space was inserted as suggested.

Generally speaking, the proposed paper in my opinion presents an interesting work and is well-written.

Thank you.

English language was also revised and improved.

Round 2

Reviewer 1 Report

The authors responded and modified the text according to the request